# Developing a Contextually Appropriate Integrated Hygiene Intervention to Achieve Sustained Reductions in Diarrheal Diseases

**Tracy Morse** [1,2,*], **Kondwani Chidziwisano** [1,2,3], **Elizabeth Tilley** [2,3,4], **Rossanie Malolo** [2], **Save Kumwenda** [2,3], **Janelisa Musaya** [5] **and Sandy Cairncross** [6]

1    Department of Civil and Environmental Engineering, University of Strathclyde, Glasgow G1 1XJ, UK
2    Center for Water, Sanitation, Health and Appropriate Technology Development (WASHTED), University of Malawi (Polytechnic), Blantyre 3, Malawi
3    Department of Environmental Health, University of Malawi (Polytechnic), Blantyre 3, Malawi
4    Eawag, Swiss Federal Institute of Aquatic Science and Technology, 8600 Duübendorf, Switzerland
5    Department of Biomedical Sciences, University of Malawi (College of Medicine), Blantyre 3, Malawi
6    Department of Disease Control, London School of Hygiene and Tropical Medicine, London WC1E 7HT, UK
*    Correspondence: tracy.thomson@strath.ac.uk

**Abstract:** Diarrheal disease in under-five children remains high in Sub-Saharan Africa; primarily attributed to environmental pathogen exposure through poorly managed water, sanitation, and hygiene (WASH) pathways, including foods. This formative study in rural Malawi used a theoretical base to determine the personal, social, environmental, and psychosocial factors that are to be considered in the development of an integrated intervention for WASH and food hygiene. Using a mixed methods approach, a stakeholder analysis was followed by data collection pertaining to 1079 children between the ages of four to 90 weeks: observations ($n = 79$); assessment of risks, attitudes, norms and self-regulation (RANAS) model ($n = 323$); structured questionnaires ($n = 1000$); focus group discussions ($n = 9$); and, in-depth interviews ($n = 9$) (PACTR201703002084166). We identified four thematic areas for the diarrheal disease intervention: hand washing with soap; food hygiene; feces management (human and animal); and, water management. The contextual issues included: the high level of knowledge on good hygiene practices not reflected in observed habits; inclusion of all family members incorporating primary caregivers (female) and financial controllers (male); and, endemic poverty as a significant barrier to hygiene infrastructure and consumable availability. The psychosocial factors identified for intervention development included social norms, abilities, and self-regulation. The resulting eight-month context specific intervention to be evaluated is described.

**Keywords:** WASH; food hygiene; complementary foods; RANAS; Malawi

## 1. Introduction

Diarrheal disease continues to be the second leading cause of death in children under five, with approximately 700,000 deaths worldwide annually [1]. It is estimated that 50% of these deaths occur in Africa and 72% occur in the first two years of life, resulting in a higher mortality rate in children than HIV, tuberculosis, and malaria combined [1,2]. The WHO has continued to emphasize not only the importance of effective treatment, but also the integral role that prevention plays in the control of diarrheal disease, which highlights priorities, such as rotavirus and measles vaccinations; promotion of early and exclusive breastfeeding and vitamin A supplementation; promotion of handwashing with soap; improved water supply quantity and quality; and, community-wide sanitation promotion [3]. Despite the fact that these types of water, sanitation, and hygiene (WASH) interventions are generally

cost effective, there has been little progress in achieving implementation at scale [4]: less than 5% of the population of Sub Saharan Africa have access to combined improved water, sanitation, and hygiene, as described by the sustainable development goal indicators [4]. However, progress has been elusive as recent publications have highlighted the various technological, social and financial complexities of reducing diarrheal disease through seemingly simple interventions [2,5–9].

The WASH Benefits and SHINE studies in Kenya and Zimbabwe, respectively, reported no impact of a range of WASH interventions on the incidence of diarrheal disease, despite extensive formative research to inform and support the development of the intervention content and delivery [10–12]. However, the WASH Benefits study in Bangladesh did demonstrate a small reduction in diarrhea, albeit with evidence that there was no benefit from a combined WASH intervention over individual sanitation or hygiene programs [13]. This may be attributed to a number of factors, including the large number of pathways in which children may become exposed to diarrheal disease pathogens. Studies within the African region have demonstrated significant contamination of the environment from both human and animal feces, which may contribute to disease transmission during environmental interactions [14–19]. Studies have also demonstrated the potential role of food contamination in diarrheal disease transmission, particularly complementary foods, which have been found to have higher levels of contamination than drinking water [20–25]. The contribution of food in the transmission of diarrhea has also been supported by the 2015 WHO report, which attributed 70% of the burden of foodborne disease to sub Saharan African and South East Asia, with 40% of this affecting children under the age of five [26]. Attempts to model the complex mechanisms that potentially link poor sanitation and hygiene to diarrheal disease, enteric enteropathy, under nutrition, and child development, highlight the challenges of understanding the myriad of environmental transmission routes and sources of contamination, which may contribute to diarrheal and other related diseases [1,5,27–29].

Despite the need for understandable, applicable information regarding how to prevent diarrhea at the household level, access to knowledge alone does not achieve sustained hygiene behavior change [30]. To achieve sustained behavior change, it is essential to consider the effects and impact of all personal, social, environmental, and psychosocial factors that directly and indirectly relate to hygiene practices, including the structural and socio-economic barriers that household members may face [31]. Models to promote positive, sustained behavior change in the WASH sector must therefore have a strong element of human psychology to support knowledge and technological based interventions [30,32]. Within the WASH sector, several models, such as Risks, Attitudes, Norms, Abilities, and Self-Regulation (RANAS) [31], Behavior Centered Design (BCD) [33], and SaniFOAM [34] have been developed, and shown, to achieve this. For example, recent studies conducted in low income countries have demonstrated the potential impact of individual training, follow-up and participatory approaches (with hazard analysis principles) on the safety of domestically produced complementary foods [35–38]. Demonstrating significant reductions in the microbiological contamination of high-risk complementary foods, and achieving improved practices in food hygiene behaviors, these interventions focused on the cleanliness of utensils, hand washing with soap (HWWS), proper storage, thorough reheating and water/milk treatment, in coordination with emotional drivers and changes in behavioral settings. However, these studies did not assess the impact of these outcomes on health indicators within their settings.

Malawi has a 22% prevalence rate of under-five childhood diarrhea, which peaks at 41% between six to eleven months [39]. With multiple WASH related pathways by which under-fives might be exposed to diarrheal disease agents, any intervention must be cognizant of contextual and psychosocial factors within the target setting. As such, the aim of this research was to develop a context specific intervention to compare the health outcomes (Table 3) of household food hygiene interventions vs WASH + food hygiene interventions in rural Malawi. The intervention was developed on the basis of formative research described here. Based on our literature review, to date, no detailed assessment of personal, social psychosocial, and environmental factors that affect food hygiene practices has been conducted. Therefore, such an assessment allows for the identification of critical factors that are to be addressed by a behavior change intervention.

## 2. Methods and Results

### 2.1. Formative Research

The formative research and intervention development were undertaken in four stages (Figure 1) to ensure the final intervention trial took into consideration learning from previous studies [30–32,39], as well as ensuring that the specific context was clearly understood in terms of social, physical, personal, and psychosocial barriers and opportunities for behavior change and improved health outcomes. To achieve this, a mixed method approach was used, to provide a full picture of both knowledge and practice, thereby validating quantitative findings and supporting the iterative process of intervention design (Figure 2). This extensive formative stage then informed the intervention development and implementation mechanisms, including appropriate tools for evaluating primary and secondary outcomes.

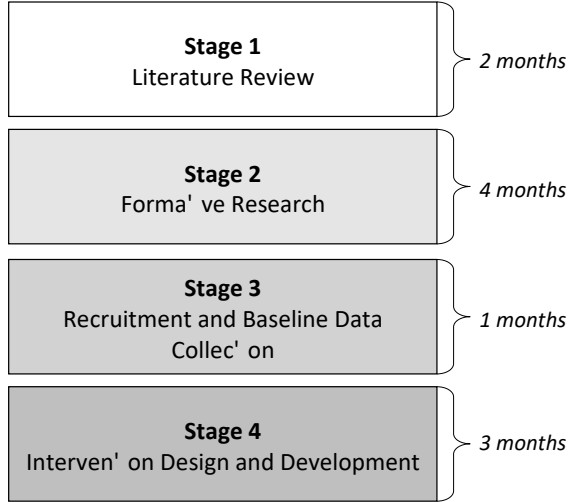

**Figure 1.** Study stages of formative research and intervention development.

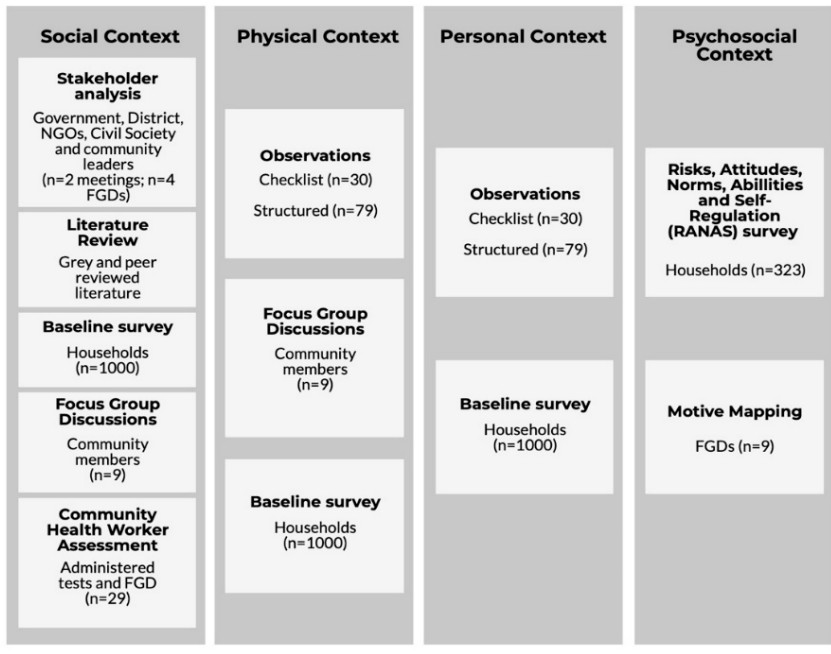

**Figure 2.** Summary of methods used to inform and undertake formative research and intervention design.

## 2.2. Setting and Population

The formative study and intervention trial both took place in the Southern Region of Malawi in Chikwawa District. Covering an area [39,40] of 4755 km$^2$, the district has an estimated population of 518,287 [41], of which 16% are under the age of five years, with an average of 4.4 people per household. The District has an under-five mortality rate of 90 deaths per 1000 births compared to 85 at national level [39]. Full vaccination coverage is 62.8%, which is slightly higher than the national average (54%), however diseases, such as diarrhea, remain higher in Chikwawa (26.3%) than nationally (22%) [39,40]. Acute respiratory infections rates among under five children are 9% (7.8% nationally). 70% of children under six months were reported to be exclusively breastfed with 88.6% being introduced to solid foods after the recommended six months. Being rural, the Chikwawa district is one of the districts with the lowest literacy rate (65.2% young female and 70.4% young male) and ranks low on the economic indicator wealth index [40]. Access to improved water sources in Chikwawa is 91.9%; however, improved sanitation coverage is 19.6%; over 40% of those with a toilet have a soil/wood floor [40]. Seven percent of households have hand washing facilities, which is half the regional average (13.8%) and only 1.3% of households have hand washing facilities with soap and water available [40]. However, 44% of households have soap available for other needs within the home [40].

## 2.3. Ethics and Consent to Participate

Ethical approval for this study was received from the University of Malawi College of Medicine Research Ethics Committee (P.04/16/1935) Trial Registration Number: PACTR201703002084166. Written informed consent and assent was obtained from all the caregivers of children participating in the study.

## 2.4. Recruitment and Participants

Malawi is divided into 28 Districts that are subdivided into Traditional Authorities (TAs). Each TA contains villages, which are administered by chiefs and/or village heads. There are 12 Traditional Authorities (TAs) within Chikwawa district. This work was based in three different TAs: one for each intervention arm and one for control. We selected the TAs in collaboration with the District Coordinating Team (inter sectoral team). Formative research took place in the same TA as the intervention, but amongst non-enrolled households (Figure 3).

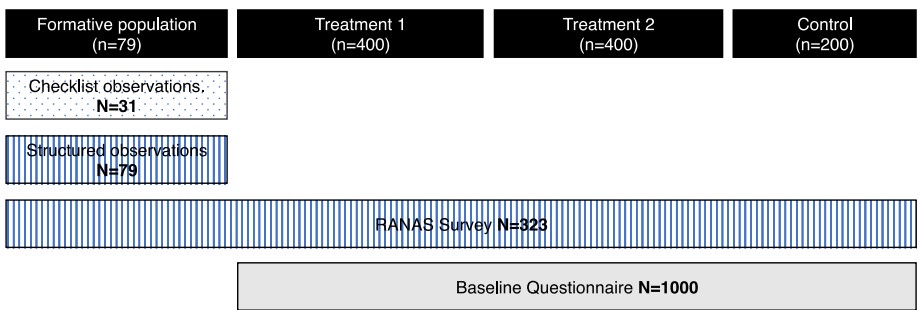

**Figure 3.** Sampling framework for formative and baseline stages.

All of the recruited households (both formative and intervention) had a functioning latrine and resided within a 500m radius of a functioning borehole to ensure that there were no significant variations in access to water or sanitation infrastructure. Eligible households had a child that was aged between four and 90 weeks at enrolment to ensure that children were not neonates and that all children were under 60 months at the end of intervention period. The age of children was verified through birth and/or immunization records supplied by the caregiver. All the children in the target age range from eligible households were recruited at baseline, however only one child per household could be recruited. Physical recruitment was conducted by trained research assistants with the approval and support of community health workers (health surveillance assistants), traditional leaders (village

chiefs), and community volunteers. Written consent was received from all households that were willing to participate before allocation of a household identification number and associated barcode.

## 2.5. Literature Review and Key Informant Interviews

A literature review was conducted to fully understand preceding food hygiene studies, their theoretical basis, methodologies, and relative outputs in term of health, behavioral, and food quality indicators. This process identified key studies and methods from which to build the formative research methodologies [25,31,33,37,42,43], and an overview of the policy landscape within the country at that time. The findings of the literature review were used in the development of an effective formative research stage, and the development of a more detailed stakeholder analysis, which was used to provide country and district specific information.

The national and district stakeholder analysis was conducted through a national meeting ($n = 65$ participants), a district executive committee meeting ($n = 25$ participants), and focus group discussions with district and non-governmental staff ($n = 2$) that were working in the study area, and community leaders ($n = 2$). These face to face meetings were conducted as public discussion forums (national and district level), and focus group discussions (district and community level). The participants were presented with the findings of the literature review and an overview of the proposed research, which they were then asked to comment on, based on previous experience and current knowledge. Information was also gathered on current programs and organizations working in the same geographical or subject areas, which were then followed up. All dialogue from the stakeholder meetings was subject to thematic analysis. With specific reference to the hygiene of complementary foods, stakeholders highlighted the issue of food safety as an 'implied concept' within WASH and nutrition programs, rather than explicit in its content and delivery. It also underlined the challenges in prioritizing diarrheal disease due to the lack of specific policy or strategy addressing this disease [44]. Lastly, concerns were raised regarding the sustainability of interventions in the WASH sector, and how a specific program should ensure that sustained behavior change is at the core to improve the health and well-being of the target population.

## 2.6. Observations

Checklist and structured observations were used for the identification of critical control points, followed by in-depth interviews. Initially, checklist observations were conducted in 30 randomly selected households in formative research households to provide a list of behaviors that were considered to put children at risk of developing diarrhea. For the checklist observations, a household was visited over two consecutive days: 6 am–12 pm on the first day and 12 pm to 6 pm on the second. The aim was to capture all the events of interest that occurred in a day. Subsequently, structured observations were conducted, specifically focusing on behaviors that were noted during checklist observations. In total, 79 households were targeted for structured observations and they were visited once from 7 am to 1 pm, as checklist observations had informed that the majority of food preparation and feeding events took place in the morning. In-depth interviews followed each checklist and structured observation period to understand how and why some practices were conducted as observed. The data was analyzed while using content analysis. Common risky practices noted during checklist observations that were further observed during structured observations were: child defecation; adult defecation; poor hand washing (after latrine use, after cleaning child's bottom, before food preparation, before child feeding/eating, before breastfeeding); water management (source management, collection, use, and storage); animals and their feces in the compound; purchase, storage, and consumption of raw food (including fruits); preparation and storage of cooked food; reheating of left-over food; washing and storage of utensils; and, child feeding. Upon further analysis of structured observation data, the following practices were selected as the critical areas for control:

- HWWS at critical times (i.e., after latrine use, after cleaning child bottom, before food preparation and before child feeding/eating).

- Food hygiene: washing utensils with soap, keeping utensils and cooked food on an elevated place, reheating of left-over food until boiling and hygienic feeding.
- Feces management: safe and immediate removal of child and animal stools from the household compound into the latrine.
- Household water management: regular washing of water collection and storage containers with soap and covering of water during storage with tight fitting lids.

## 2.7. RANAS Questionnaire and Motive Mapping

Following the observation stage, each critical hygiene point was examined to identify the psychosocial factors for the selected behaviors using a RANAS model-based [31] household questionnaire. To our knowledge, a RANAS model has not been previously applied in a food hygiene assessment. However, it has been successfully used to evaluate and achieve behavior change in water treatment, sanitation, and hand washing behaviors, and it offers a clear and structured process for data collection, analysis and identification of potential behavior change techniques to use in subsequent interventions [45–48].

The questionnaire captured demographics, hygiene behaviors, socio-economic proxy measures, and psychological variables of the RANAS model for the targeted behaviors (Table 1). A household wealth index to assess socio-economic status was measured while using principal component analysis. Behavioral factors were measured through targeted questions for specific behaviors. Information regarding health and practices were measured with multiple response questions from which the enumerators chose answers given by the respondent. Behavioral factors for critical hygiene points of interest were assessed using unipolar responses ranging from 1 to 5. Statistical analysis of the behavioral factors was conducted using IBM SPSS software version 25. Only behavioral factors that significantly correlated with dependent variables in ANOVA analysis were included in the Cohen's D test analysis to identify critical factors to be targeted with an intervention.

The skills of community health workers ($n = 29$) were also evaluated to determine their knowledge and awareness of the critical hygiene controls identified in the formative study. This was undertaken through a supervised written assessment that was performed on a one to one basis and subsequent FGD ($n = 2$) to explore the key findings.

Following the identification of the key behavioral factors (Table 1), nine focus group discussions were conducted ($n = 6$ women; $n = 3$ men). These were used to validate the findings of the observations and psychosocial factors, and included a motive mapping session, which was used to assess immediate motives for the key behaviors. Seven different pictures demonstrating common motives for each behavior (attraction, nurture, disgust, status/respect, affiliation, purity, and disease) were shown and the stories were articulated around those pictures to identify the emotional drivers. Caregivers were then asked to rank the pictures according to how likely these would motivate them to practice key behaviors.

## 2.8. Baseline Data Collection

Baseline and demographic data were collected from all households at recruitment (Figure 2). A structured survey was conducted to collate the information pertaining to: household membership; house construction; sanitation and hygiene facilities; water access, source and storage practices; child health; child food and feeding practices; and, animal ownership.

All the data were collected on tablets while using Open Data Kit (ODK) and sent for collation and cleaning before analysis in version STATA 13.1. Table 2 summarizes the results.

**Table 1.** Summary of behavioral factors and motives identified as significant for inclusion in the intervention.

| Targeted Behavior | Sub Item | Group Factor | Behavioral Factors | Motivational Drivers |
|---|---|---|---|---|
| Hand washing with soap at critical times: | • Before food preparation<br>• Before child feeding/eating<br>• After changing baby nappy<br>• After latrine use | Norm | • Others practicing behavior | Status and affiliation |
| | | Ability | • Confidence in performance (HW facility)<br>• Confidence in performance (soap)<br>• Confidence in continuation (time) | |
| | | Self-regulation | • Remembering (forgetting)<br>• Remembering (attention)<br>• Commitment (committed) | |
| | | Attitude | • Feeling | |
| Food hygiene | • Washing utensils with soap | Ability | • Confidence in performance (commitment)<br>• Confidence in continuation | Affiliation, purity, nurture and attraction |
| | | Attitude | • Emotion (pleasant) | |
| | | Norm | • Others practicing behavior<br>• Others approval | |
| | • Keeping utensils on elevated place | Risk | • Vulnerability | |
| | | Norm | • Others practicing behavior | |
| | • Reheating of left-over food | Ability | • confidence in performance<br>• Confidence in continuation<br>• Confidence in recovery | |
| | | Attitude | • Pleasant | |
| | | Norms | • Others practicing behavior<br>• Personal importance | |
| | | Abilities | • Confidence in performance (because of inadequate firewood)<br>• Confidence in continuation (because of inadequate firewood)<br>• Confidence in recovery ((because of inadequate firewood) | |
| | • Child feeding practices | Attitude | • Pleasant<br>• Like | |
| | | Norms | • Others practicing behavior | |
| | | Ability | • Confidence in performance (they don't have time) | |
| Animal and child feces management | • Immediate removal of child feces with proper materials<br>• Regular sweeping of household environment | Risk | • Vulnerability | Disgust, diseases and status |
| | | Norm | • Others practicing behavior<br>• Personal obligation | |
| | | Ability | • Confidence in performance (difficult)<br>• Confidence in performance (hurry) | |
| Household water management | • Animals accessing water for domestic purposes | Risk | • Vulnerability | Affiliation, purity, diseases and disgust |
| | | Attitude | • Feeling | |
| | • Containers for collecting and storing water not washed with soap | Risk | • Vulnerability | |
| | | Norms | • Others practicing behavior | |
| | • Water storage containers not covered, or covered with untighten covering lid | Ability | • Confidence in performance | |

**Table 2.** Descriptive statistics for 1000 households from the baseline survey (For description of Treatments 1 and 2, see below).

| | Treatment 1 | Treatment 2 | Control |
|---|---|---|---|
| N (unless noted otherwise) | **400** | **400** | **200** |
| Female child | 0.49 | 0.52 | 0.55 |
| Child has a Health passport | 0.96 | 0.96 | 0.98 |
| Age distribution 1–5 months | 0.21 | 0.21 | 0.17 |
| 6–11 months | 0.28 | 0.31 | 0.31 |
| 12–17 months | 0.32 | 0.31 | 0.37 |
| 18+ months | 0.19 | 0.18 | 0.16 |
| **Health status** | | | |
| *Vaccinations* | | | |
| BCG (TB) vaccination at birth | 0.97 | 0.98 | 0.99 |
| Oral Polio Vaccine (OPV) at birth | 0.91 | 0.89 | 0.98 |
| % of eligible children at 6 weeks receiving OPV; Diphtheria, Pertussis, Tetanus, Hepatitis B, Influenza B | 0.96 | 0.97 | 0.98 |
| % of eligible children at 10 weeks receiving OPV; Diphtheria, Pertussis, Tetanus, Hepatitis B, Influenza B | 0.93 | 0.94 | 0.98 |
| % of eligible children at 14 weeks receiving OPV; Diphtheria, Pertussis, Tetanus, Hepatitis B, Influenza B | 0.87 | 0.84 | 0.89 |
| % of eligible children at 6 months receiving Vitamin A | 0.78 | 0.68 | 0.80 |
| % of eligible children at 9 months receiving Measles, Mumps and Rubella | 0.87 | 0.77 | 0.93 |
| % of eligible children at 12 months receiving Vitamin A | 0.68 | 0.55 | 0.61 |
| % of eligible children at 18 months receiving Vitamin A | 0.71 | 0.56 | 0.58 |
| *Illness in last 2 weeks* | | | |
| Child had diarrhea in the last 2 weeks | 0.45 | 0.45 | 0.43 |
| *Child was treated* | 0.35 | 0.34 | 0.24 |
| Children <6 months | 0.21 | 0.21 | 0.17 |
| *Who had diarrhea in the last 2 weeks* | 0.13 | 0.16 | 0.18 |
| Children 6–11 months | 0.28 | 0.31 | 0.31 |
| *Who had diarrhea in the last 2 weeks* | 0.54 | 0.54 | 0.48 |
| Children 12–23 months | 0.51 | 0.49 | 0.52 |
| *Who had diarrhea in the last 2 weeks* | 0.53 | 0.51 | 0.47 |
| Child had respiratory infection in last 2 weeks | 0.56 | 0.62 | 0.57 |
| *Child was treated* | 0.43 | 0.48 | 0.34 |
| Child had skin infection in last 2 weeks | 0.18 | 0.18 | 0.20 |
| Child had eye infection in last 2 weeks | 0.05 | 0.04 | 0.04 |
| Child had ringworm in last 2 weeks | 0.02 | 0.01 | 0.00 |
| Child had schistosomiasis in last 2 weeks | 0.01 | 0.00 | 0.00 |
| **Child Feeding** | | | |
| Child is breastfeeding | 0.98 | 0.99 | 1.00 |
| Among breastfeeding children, child drinks water | 0.86 | 0.89 | 0.86 |
| Child eats solid food | 0.82 | 0.85 | 0.85 |
| Number of meals child had yesterday (N = 799) * | 2.22 | 2.12 | 2.13 |
| **Sanitation and Hygiene** | | | |
| Where nappies are disposed of | | | |
| *Household yard* | 0.1 | 0.15 | 0.19 |
| *Pit latrine* | 0.82 | 0.77 | 0.65 |
| *Various* | 0.03 | 0.03 | 0.07 |
| *Other* | 0.05 | 0.06 | 0.10 |
| Flies are visible | 0.59 | 0.55 | 0.51 |
| Hand washing facility exists | 0.36 | 0.18 | 0.23 |
| Of those with facility, facility type | | | |
| *Basin + jug* | 0.16 | 0.38 | 0.09 |
| *Handmade from bottle* | 0.79 | 0.56 | 0.91 |
| *Other* | 0.05 | 0.07 | 0.00 |
| Household has soap | 0.61 | 0.59 | 0.63 |
| Of households with soap | | | |
| *Used for bathing* | 0.65 | 0.60 | 0.70 |
| *Used for clothes washing* | 0.82 | 0.81 | 0.82 |
| *Used for washing hands* | 0.62 | 0.56 | 0.47 |
| *Used for washing kitchen utensils* | 0.46 | 0.53 | 0.56 |
| **Drinking water** | | | |
| Drinking water storage | | | |
| *Metal bucket* | 0.00 | 0.00 | 0.00 |
| *Plastic bucket* | 0.24 | 0.37 | 0.38 |
| *Jerrycan* | 0.22 | 0.32 | 0.30 |
| *Clay pot* | 0.52 | 0.26 | 0.25 |
| *Other* | 0.02 | 0.06 | 0.09 |
| *Average total volume of water storage (L) * | 98 | 117 | 101 |
| Water storage method | | | |
| *not covered* | 0.14 | 0.13 | 0.25 |
| *cover* | 0.14 | 0.13 | 0.12 |
| *total cover* | 0.72 | 0.72 | 0.58 |
| *not observed* | 0.00 | 0.02 | 0.05 |
| **Animal ownership** | | | |
| Number of animals that the household owns * | 1.43 | 1.44 | 1.20 |

\* indicates that the result is a value and not a percentage.

Overall, the households were primarily constructed from mud (50%) or burnt bricks (42%), and the toilets were traditional pit latrines with soil floors (96%). Complementary foods being consumed by children included maize and sorghum porridge, which in some cases included groundnut flour.

The consolidation of the baseline results with the findings of observations and RANAS, identified key contextual issues that need to be considered in the development of an effective intervention. Specifically:

- Although the mother was identified as the primary caregiver in this rural context, there is a need for strong male involvement in the intervention, as they are the primary managers and decision makers in terms of household finance.
- There was marked homogeneity across all households in terms of diet, cooking, food storage, feeding practices, and available utensils. This allows us to design an intervention to easily tackle food safety issues in a generic manner across the population.
- Comparison of data from baseline and observations demonstrated that knowledge of good hygiene practice is high. However, this does not translate into practice, and as such the intervention should have a behavior-centered approach to address this gap (Figure 4).
- Caregivers play a number of roles within their household and community, and any intervention needs to be cognizant of time commitments and availability.
- Although the results showed that health workers had a good level of understanding of water and sanitation issues (80% know to use water and soap for effective hand washing; 93% believe poor HWWS due to lack of soap/poverty; 69% understood animal feces could transmit disease), there was limited knowledge in relation to food hygiene practices (10% could describe critical behaviors around food hygiene). This needs to be addressed if they are to facilitate the intervention.
- The intervention must be cognizant of the low level of literacy and the level of poverty within the target population, and therefore behavior change techniques must be appropriate, and recommendations need to be realistic.

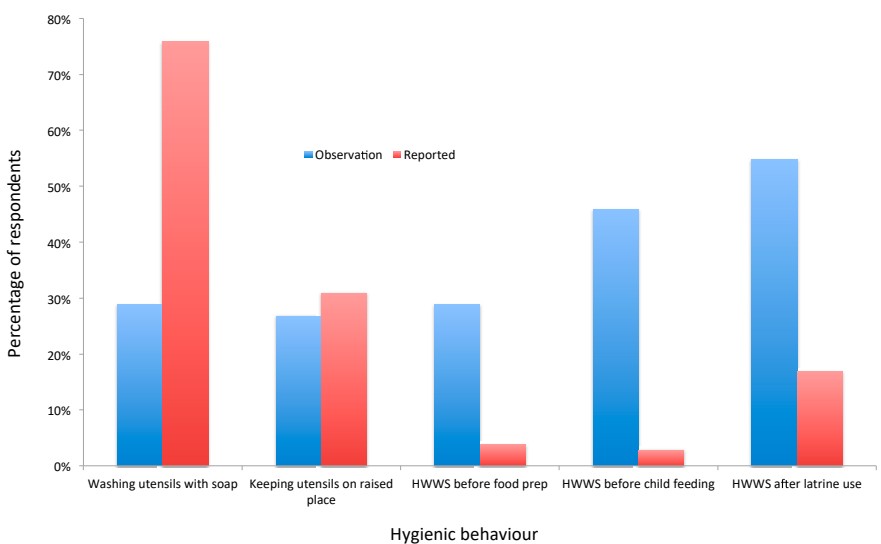

**Figure 4.** Self-reported behaviors in households versus those observed during checklist and structured observations.

## 3. Intervention Design

The intervention was designed as a randomized cluster before and after trial with a control, and two treatment arms (Figure 5). The inclusion of two treatment arms was to measure the relative impact of the hygiene of weaning foods on primary and secondary outcomes (Table 3). Therefore, this would allow us to measure:

- The impact of each intervention on the incidence of diarrheal disease.
- The impact of each intervention relative to the Control Group.
- The impact of the WASH intervention relative to the WASH + Food Hygiene Intervention.

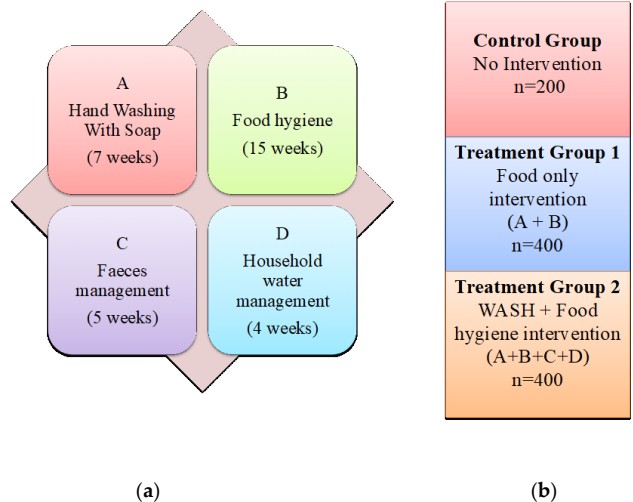

(**a**)  (**b**)

**Figure 5.** (**a**) Intervention design showing the key behaviors to be addressed and (**b**) the breakdown of intervention delivery by treatment arm and population.

**Table 3.** Variables to be measured to determine relative impact of interventions on primary and secondary outcomes.

| Outcome | Variable | Method of Measurement | Point of Measurement |
|---|---|---|---|
| Primary | Diarrheal disease | Self reporting<br>Household/Cluster checks with community volunteer<br>Community health worker reports | Continuous<br>Baseline (illness 2 weeks prior)<br>End line (illness 2 weeks prior) |
| Secondary | Health<br>• Eye infections<br>• Acute respiratory infections | Self reporting | Continuous<br>Baseline (illness 2 weeks prior)<br>End line (illness 2 weeks prior) |
| | Behavioral factors and changes | Risks, Attitudes, Norms, Abilities and Self Regulation questionnaire<br>Observations | Baseline<br>Midline<br>End line |
| | Changes in household environment | Checklists<br>Observations | Fortnightly checklists<br>Baseline<br>End line |
| | Microbiological contamination of<br>• Food<br>• Environment<br>• Hands (critical times)<br>• Stools (human & animal) | Microbiological testing for:<br>• Aerobic plate count<br>• Faecal coliforms<br>• *E. coli*<br>• *Salmonella*<br>• *Shigella*<br>• *Staphylococcus aureus*<br>• Soil transmitted helminthes<br>• Protozoa (*Crytosporidium* & *Giardia*)<br>• Rotavirus | End line 100 households per treatment arm and control |
| | Non-WASH benefits | Community networks analysis<br>Social capital measurement | End line |

The intervention took into consideration the findings of our formative research, baseline data, and the lessons of previous studies [10,13,35–38]. As such, the intervention was named "The Hygienic Family" (*Banja la Ukhondo*) to ensure the inclusion of all family members and support the concept of whole community improvements versus individualism.

The intervention framework was based on the critical points that were identified within the formative research, which were consolidated by the research team under four thematic areas to form intervention packages (Figure 3). Each of these critical points was then examined in terms of context (social, personal, and environmental), structural barriers, and psychosocial factors (RANAS model) to design specific intervention activities. These activities were then developed with an in-house design team to produce specific and complementary modules, which could be delivered through community-based volunteers with support from community health workers (Table 4).

**Table 4.** Summary of intervention packages.

| Intervention Package | Event | Purpose | Key Content |
|---|---|---|---|
| **Handwashing with soap (7 weeks)** | First to fourth cluster meeting | • Present situations in everyday life of the caregivers, practically showing how unhygienic handwashing behavior leads to diarrheal disease.<br>• Increase caregivers' confidence by paying attention to others' performing the behavior.<br>• Describe feelings about performing and about consequences of the behavior in their everyday life.<br>• Prompt and support the caregivers to set up handwashing facilities. | • Demonstration diarrhea pathway using fecal oral route illustration.<br>• Paint game demonstrating disease spread.<br>• Hand washing with soap commitment-Paper plate hand painting.<br>• Distribute hand washing with soap score cards.<br>• Guided practice on steps to washing hands with soap.<br>• Demonstrate and discuss difference in feelings between handwashing with and without soap.<br>• Singing handwashing with soap song.<br>• Hand washing with soap Glo Germ Experiment.<br>• Benefits of hand washing with soap-Video.<br>• Reward the performers of handwashing with soap. |
| | First to third household visit | • Reinforce correct hand washing with soap practice at four critical times, with the view to these becoming habitual. | • Provide one to one guided practice about handwashing with soap.<br>• Caregiver's handwashing observed and corrected where necessary.<br>• Identifying performers by observing existing practices.<br>• Provide one to one practical guidance on handwashing facility construction (i.e., tippy tap), and location (e.g., latrine and cooking area). |
| **Food hygiene (15 weeks)** | First to eighth cluster meeting | • Enhance confidence in performance (continuation)—empower caregivers not to forget hand washing with soap at four critical times.<br>• Create affiliation and habit formation (washing utensils with soap, keep utensils on raised place, reheating food, child feeding).<br>• Build confidence in performance: others support and perform the behaviors (use of role models).<br>• Reinforce habit formation about handwashing with soap, washing utensils with soap, keep utensils on raised place, reheating food, child feeding.<br>• Establish reheating of left-over food and safe food storage as a social norm. | • Paint game to enhance handwashing with soap, distribute bracelets to act as handwashing reminders, each caregiver receives a certificate to indicate and acknowledge their commitment in hand washing with soap.<br>• Puzzle game to initiate habit formation about washing utensils with soap, group norms elicited by role models, washing utensils with soap demonstration.<br>• Cooking demonstration to motivate handwashing before food preparation, washing utensils with soap and handwashing with soap before feeding,<br>• Card game.<br>• Child feeding demonstration.<br>• Cardboard shuffling game-keeping utensils on a raised place.<br>• Demonstration of dish rack construction and caregivers commitment to own and use dish racks. Supported by subsequent dish rack construction awards.<br>• Role models to motivate others about caregivers feeding their children. Practical session about consequences of poor child self–feeding, distribute bibs and buntings as cues.<br>• Pass the ball game to demonstrate how food stuffs and leftovers are stored.<br>• Role play to promote reheating of food.<br>• Fixing my food and utensil storage area competition. |
| | First to seventh household visit | • Prompt washing utensils with soap and keeping utensils on raised place practices.<br>• Reinforce reheating of left-over food and safe child feeding practice. | • Observe caregiver's washing utensils with soap, keeping utensils on raised place and handwashing with soap at critical times, and corrected where necessary.<br>• Point out the pleasant feeling a caregiver gets when they always feed children themselves, with use of flip illustrations to motivate caregivers to always feed their children.<br>• Observe and encourage use of bracelets, bibs and buntings.<br>• Provide practical support on dish rack construction.<br>• Using illustrations, encourage caregivers on the importance of using dish racks.<br>• Demonstrate good storage practice of left-over food. |

**Table 4.** *Cont.*

| Intervention Package | Event | Purpose | Key Content |
|---|---|---|---|
| **Child and animal feces management (5 weeks)** | First to third cluster meeting | • Remind caregivers and create disgust about the risk of delaying removal of feces with unrecommended materials and irregular sweeping of household environment.<br>• Encourage role models to remind caregivers that others already perform the behavior.<br>• Ensure that safe removal of child and animal feces including cleaning of environment is integrated into caregiver's daily activities. | • Create disgust: Feces eating game-water, feces eating game-food,<br>• Role play-proper removal of feces,<br>• Picture cards game.<br>• Role models demonstrating how they achieve clean surroundings,<br>• Correct feces removal demonstration and prompt other caregivers to pay attention to others' performing the behavior and its consequences in their everyday life,<br>• Step by step feces removal process poster discussion.<br>• Hygienic family poster discussion,<br>• Drama—to inform about others already performing the behavior,<br>• UNICEF Food Hygiene videos, Local chief commitment.<br>• Receive posters about feces removal process and *Banja la Ukhondo*, (hygienic family)<br>• Reward performers of good practice. |
| | First to second household visit | • Reinforce correct child and animal feces behavior.<br>• Reinforce previous oriented behaviors related to handwashing with soap and food hygiene practices. | • Observe if the surrounding is clean (i.e., free of child and animal feces) and correct where necessary<br>• Observe promote use of bracelets, bibs and buntings.<br>• Demonstrate proper removal and disposal of feces using the feces removal process flow chart |
| **Household water management (4 weeks)** | First to second cluster meeting | • Remind caregivers and create disgust about the risk of not covering water for drinking and other domestic purposes with a tight fitting cover.<br>• Enhance confidence in performance (continuation): empower caregivers to always wash water collection and storage containers with soap.<br>• Create affiliation and habit formation about covering water containers with tight fitting cover and also wash water collection and storage containers with soap. | • Create disgust: Feces eating game-water, feces eating game-food,<br>• Practical demonstration on how water that is clean gets contaminated at the household<br>• Picture discussion how animals contaminate water for drinking and for other domestic purposes when at household.<br>• Use of glo-germ to demonstrate unpleasant feelings about performing and about consequences of allowing animals to access stored water for domestic purposes<br>• Use of cholera story video to describe feelings about performing and about consequences of not covering water with tight fitting cover and also not cleaning water collection and storage containers with soap. |
| | First to second household visit | • Reinforce covering of water with tight fitting cover and cleaning of water collection and storage containers with soap behaviors.<br>• Reinforce previous oriented behaviors related to handwashing with soap, food hygiene practices and feces management. | • Observe if water storage containers are covered with tight fitting covers and not easily accessed by animals. Provide support where necessary.<br>• Observe if water collection and storage containers are cleaned with soap. Provide support where necessary.<br>• Observe if the surrounding is clean (i.e., free of child and animal feces) and correct where necessary<br>• Observe promote use of bracelets, bibs and buntings. |

The basis of delivery was derived from both the formative work and experience from previous community-based studies in Malawi [43,49,50]. These studies have highlighted the effective use of women's cluster groups, contextualized dramas, and songs in the delivery of health promotion through the leveraging of social capital and collective efficacy. The RANAS and motive mapping also informed the development of appropriate environmental prompts to support concerns that are related to self-regulation and remembering. As such, the final delivery mechanisms were agreed as: cluster group meetings; household visits; open days and public celebrations; and, communication tools, such as posters, bracelets, bibs, etc. (Table 4).

### 3.1. Sample Size

Following Rutterford et al., the population of each treatment arm (m) was calculated based on the formula [51].

$$m = \frac{\left(Z_{1-\alpha/2} + Z_{1-\beta}\right)^2 [P_1(1 - P_1) + P_2(1 - P_2)]}{\Delta^2} \times [1 + (n - 1)\rho] \tag{1}$$

where the proportion of diarrhea ($P_1$) in the Control Area was based on the value determined during baseline data collection (0.45). The ability to detect a 15 percentage point change (i.e., 30 percent prevalence) with a power of 0.8 an alpha value of 0.05, an intra-cluster correlation coefficient (ICC) of 0.05 (assumed), a cluster size of 20 (*n*) yielded a population of 311 per treatment arm. When considering attrition, the two treatment arms were increased to a sample size of 400 (20 clusters of 20), while logistics and budgetary constraints meant that the Control sample size was limited to 200 (10 clusters of 20).

The sample size was calculated while using estimates that come from the 2016 Malawi Demographics and Health Survey (MDHS) (National Statistical Office & ICF Marco, 2016). Each Treatment group includes 20 clusters, and the control group has 10 clusters, which are the unit of measurement; diarrhea incidence is the outcome variable of interest from each of the clusters. Each cluster includes 20 children under five (separate households). Based on this sample size, we will be able to test the hypothesis that a given intervention has a statistically significant impact on the incidence of diarrhea. Overall, 1000 individuals (children under 5) will be included in the study.

### 3.2. Intervention Framework

Based on the findings of the formative and baseline data, the intervention was developed over the four critical areas, as outlined in Table 4, to take place over and intensive 32-week period (Figure 3). Cluster meetings and household visits took place on alternating weeks, being facilitated by community volunteers and supported by community health workers and intervention staff. Intervention content and delivery mechanisms were reviewed by a stakeholder research advisory group and pre-tested with the community volunteers. A critical review was also undertaken at the end of each module as part of the process evaluation.

### 3.3. Data Management and Analysis Plan

The use of mobile phones will allow for the majority of the data collected to be automatically merged into an online database. Each household was assigned a unique household identification number at baseline, which will be used to merge descriptive, observational, and microbiological data from the baseline to the end line including any and all continuous data.

Data Analysis

The primary variable of interest is the occurrence of diarrhea within the last two weeks, as reported by the primary caregiver and measured as a binary outcome. The impact of the interventions will be analyzed while using a difference-in-differences approach such that the hypothesized reduction in diarrhea in the treatment areas will be measured between the baseline and the follow-up surveys and compared to the same time points in the control areas (*n* = 1000). Attendance at cluster meetings (individual level), cluster attendance score (cluster level), socio-economic characteristics (individual level), and household hygiene improvements (e.g., installation of a dish rack) (individual level) will be included as covariates in the model. The incidence of respiratory infection and eye infections will be analyzed likewise.

Microbial contamination (Table 3) will be measured at key points of contamination, as identified during the formative research, and it will be assessed at the end of the intervention in both the treatment and control areas (*n* = 300 households for all sampling areas). The same covariates will be regressed

onto the log-transformed values to determine the predictors of various points of contamination for each of the organisms identified.

Behavior change will be analyzed by comparing the factors that were measured in the RANAS survey between the treatment and control areas between the three time points: baseline (*n* = 323), midline (*n* = 1000), and endline (*n* = 1000).

## 4. Discussion

The formative results highlight the importance of understanding context in the development of an intervention in terms of personal, social, environmental, and psychosocial factors. In particular, the results highlighted the specific structural barriers to potential intervention success, such as poverty, social norms, and concerns pertaining to household abilities and self-regulation to maintain hygienic behaviors.

These findings have been integrated into an intervention plan, which builds upon successes of previous studies in food hygiene promotion [35–38], but is also cognizant of the cultural setting of rural Malawi. This is particularly relevant to the methods that were used for intervention delivery [43,49,50].

Baseline characteristics demonstrated satisfactory randomization between the cluster and treatment/control arms. Descriptive statistics from the baseline were also indicative of the characteristics within the study site, as described in Malawi's Integrated Household Survey [39].

The intervention evaluation has been designed to measure primary health outcomes, which previous studies have not addressed. In addition, we include several secondary outcomes while using mixed methods, to support an in-depth analysis of the personal, social, environmental and psychosocial changes that the intervention might have imparted, including those not associated with specific WASH benefits.

The intervention has been designed to minimize limitations. Each Treatment Area and Control Area will be separated by a Traditional Authority to ensure that there are several kilometers and several communities between each group. The spacing of the areas is necessary for reducing spillover effects (i.e., information travelling between different Treatment Areas or from a Treatment Area to a Control Area), which could negatively impact the results.

Data collection will occur during several important seasonal changes, which are likely to affect access to water and hygiene practices (i.e., rainy and dry seasons). However, because the data that were collected in the Control Area will also be affected by seasonal changes, the final impact measured in the Treatment Areas will not be biased by these natural variations, as we will take the baseline and final values in both areas into account.

## 5. Conclusions

We present a summary of the findings of an extensive formative study assessing childcare practices that are focused on water, sanitation, and hygiene, including the hygiene of complementary foods. The findings of the literature review, stakeholder analysis, observations, baseline data collection, and psychosocial assessment have formed the basis of an intervention framework, as described here. This intervention is now being evaluated to determine its relative impact on the reduction of diarrheal disease in children under the age of five years in rural Malawi.

**Author Contributions:** All authors contributed to design of the study protocol and helped with drafting the manuscript. The following specific contributions apply: T.M., K.C., E.T. and S.C. developed the research concept and methodology. T.M., K.C., E.T., S.K., J.M. and R.M. developed data collection tools and supervised data collection. E.T. supervised data management. K.C., E.T. and T.M. undertook data analysis and interpretation. T.M., K.C. and R.M. developed the intervention content. All authors contributed to the manuscript.

**Funding:** This research and publication were made possible with UK Aid from the Department of International Development (DFID) as part of the Sanitation and Hygiene Applied Research for Equity (SHARE) Research Consortium (http://www.shareresearch.org). However, the views expressed do not necessarily reflect the Department's official policies.

**Acknowledgments:** The authors gratefully acknowledge the support of SHARE Consortium management team in the development and successful implementation of this formative research. We also acknowledge the support provided by Hans Mosler and Jurgita Slekiene EAWAG (Switzerland) for their support in the development and analysis of the context specific RANAS questions. A special thanks to all government extension workers, field data collectors, volunteers and household members who participated in the study.

**Conflicts of Interest:** The authors declare that they have no competing interests.

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
