# Peer review of "Developing a Contextually Appropriate Integrated Hygiene Intervention to Achieve Sustained Reductions in Diarrheal Diseases"

_sustainability, doi:10.3390/su11174656_

Round 1

Reviewer 1 Report

The article described a formative study that was used to inform the design of a complex intervention aimed at reducing the incidence of diarrhoeal disease in children under the age of five in rural Malawi. The study was large, well designed and interesting. I have a few comments/suggestions for Authors:

1) Some editing and reformatting is required. The tables were a little difficult to read as contents were aligned centre, rather than left. A number of the figures also appeared to have text formatting errors. There were a few minor typographical errors. Some of the text in section 2, esp lines 10 - 99, appeared to have a reasonably high similarity to previously published work (Tumwebaze, I. K., & Mosler, H. J. (2014). Shared toilet users' collective cleaning and determinant factors in Kampala slums, Uganda. BMC public health14, 1260. doi:10.1186/1471-2458-14-1260). Please consider rephrasing some of this section.

2) I wondered if eye infection incidence was too low to be a meaningful secondary outcome measure. Perhaps acute skin infection could be considered instead.

3) What were the literacy rates of adults of the treatment and control groups? Would literacy rates have any influence on the effectiveness of the interventions?

4) The Authors noted that the RANAS model had not been previously applied in a food hygiene assessment. The justification for using this model could have been stronger.

Author Response

Response to Reviewer 1: Sustainability 576431 - Developing a Contextually Appropriate Integrated Hygiene Intervention to Achieve Sustained Reductions in Diarrheal Diseases

Many thanks for your time and efforts to review our manuscript, and your useful feedback which we have used to improve the content. Specific responses to your queries are outlined below.

The article described a formative study that was used to inform the design of a complex intervention aimed at reducing the incidence of diarrhoeal disease in children under the age of five in rural Malawi. The study was large, well designed and interesting.

We appreciate your positive feedback on the overall structure of the study and intervention.

I have a few comments/suggestions for Authors:

Some editing and reformatting is required. The tables were a little difficult to read as contents were aligned centre, rather than left.

This issue has now been addressed for all tables. Initial submission was left hand justified so this may be a journal formatting issue which changed to cantered formatting. As authors we agree with the reviewer that left hand justified makes the tables more user friendly and hope that this can be maintained should the manuscript be accepted for publication,

A number of the figures also appeared to have text formatting errors. 

As your comments were not specific here we have carefully reviewed all figures and made the following changes:

Figure 1 – The title of the figure has been updated to be reflective of the content.

Figure 2 - We have reformatted Figure 2 for ease of reading

There were a few minor typographical errors.

We have reread and edited the document to identify and correct any typographical errors

Some of the text in section 2, esp lines 10 - 99, appeared to have a reasonably high similarity to previously published work (Tumwebaze, I. K., & Mosler, H. J. (2014). Shared toilet users' collective cleaning and determinant factors in Kampala slums, Uganda. BMC public health14, 1260. doi:10.1186/1471-2458-14-1260). Please consider rephrasing some of this section.

We are unsure as to the specific section referred to here by the reviewer as this covers the whole abstract and introductory section and not section 2. We have reviewed the content and are confident that this section has been prepared by the authors and does not replicate that of the paper mentioned. Please clarify section if concerns are still there.

2) I wondered if eye infection incidence was too low to be a meaningful secondary outcome measure. Perhaps acute skin infection could be considered instead.

Thank you for this observation and admittedly this will be a limitation of our study in terms of secondary outcomes. However, as reduction in eye infections is indicative of improved child care hygiene practices, and trachoma is endemic in the study area we feel this is still a valuable indicator to maintain as a secondary outcome.

3) What were the literacy rates of adults of the treatment and control groups? Would literacy rates have any influence on the effectiveness of the interventions?

As outlined in Section 2.2 Setting and population – “Being rural, Chikwawa district is one of the districts with the lowest literacy rate (65.2% young female and 70.4% young male) and ranks low on the economic indicator wealth index40.” However, we have also added to the results section the following change to acknowledge the importance of literacy in intervention development:

“The intervention must be cognizant of the low level of literacy, and level of poverty within the target population, and therefore behavior change techniques must be appropriate, and recommendations need to be realistic. “

The Authors noted that the RANAS model had not been previously applied in a food hygiene assessment. The justification for using this model could have been stronger.

We acknowledge that a stronger justification of the use of RANAS over other models is justified and have updated the text as follows:

“Following the observation stage, each critical hygiene point was examined to identify the psychosocial factors for the selected behaviors using a RANAS model-based34 household questionnaire. To our knowledge, a RANAS model has not been previously applied in a food hygiene assessment. However, it has been successfully used to evaluate and achieve behavior change in water treatment, sanitation and hand washing behaviors, and offers a clear and structured process for data collection, analysis and identification of potential behavior change techniques to use in subsequent interventions45–48.”

Reviewer 2 Report

Thanks for allowing me to review this manuscript. I think this is a very important topic and this research will contribute to the field.

Abstract – Provides relevant information and provides in-depth (and adequate) information about the study.

Introduction section –

1) I would suggest rephrasing the first sentence. Instead of using brackets, please define diarrhea and then address complications/deaths caused by the disease. For example, Diarrheal disease is defined as ….It continues to the leading cause…

2) Line 53 – please elaborate on “preparation”. More specifically, please describe (briefly) measures taken to reduce the occurrence of disease/problem.

Methods & Results –

1) Line 102 -103 - Can you briefly describe (more) why mixed method approach was chosen ?

2) I think figure 1 needs to be formatted. Please make appropriate changes.

3) I think figure 2 needs to be formatted. Please make appropriate changes. I would like to review these figures again once formatting has been completed.

Literature Review and Key Informant Interviews –

1) I would suggest dividing this section into 2 sections. In addition to this, it is important to provide information about questions that were used for this interview. Did authors use semi structured interview guide?

2) Section 2.6 – Please include information about “structured interview”. Again, I would like to see more information about questions that were asked during interview.

3) I would also like to see more information about how data analysis was conducted for interviews noted above (see 1 & 2). Did you conduct thematic analysis, content analysis or phenomenological hermeneutics ??

Other sections are well developed.

Author Response

Response to Reviewer 2: Sustainability 576431 - Developing a Contextually Appropriate Integrated Hygiene Intervention to Achieve Sustained Reductions in Diarrheal Diseases

We thank the reviewer for their time and effort to provide constructive feedback on out submitted manuscript and provide outlines of the changes based on their recommendations below:

Thanks for allowing me to review this manuscript. I think this is a very important topic and this research will contribute to the field.

Many thanks for this acknowledgement

Abstract – Provides relevant information and provides in-depth (and adequate) information about the study.

Many thanks

Introduction section – 

1) I would suggest rephrasing the first sentence. Instead of using brackets, please define diarrhea and then address complications/deaths caused by the disease. For example, Diarrheal disease is defined as ….It continues to the leading cause…

The introductory line has been rewritten to be less cumbersome for the reader by removing the definition of diarrhea as we feel this is a well defined disease and the definition is contained within the reference provided.

As such this now reads:

“Diarrheal disease continues to be the second leading cause of death in children under five, with approximately 700,000 deaths worldwide annually1.”

2) Line 53 – please elaborate on “preparation”. More specifically, please describe (briefly) measures taken to reduce the occurrence of disease/problem.

We have reworded this section as follows to clarify the intended meaning within our introduction.

“The WASH Benefits and SHINE studies in Kenya and Zimbabwe respectively reported no impact of a range of WASH interventions on the incidence of diarrheal disease, despite extensive formative research to inform and support the development of the intervention content10–12

Methods & Results –

Line 102 -103 - Can you briefly describe (more) why mixed method approach was chosen?

A fuller description to support the use of a mixed method approach has been provided as follows:

“To achieve this a mixed method approach was used, to provide a full picture of both knowledge and practice thereby validating quantitative findings, and supporting the iterative process of intervention design (Figure 2)”

I think figure 1 needs to be formatted. Please make appropriate changes.

The heading for figure one has now been updated to

Figure 1. Study stages of formative research and intervention development.”

To reflect more effectively the information contained.

I think figure 2 needs to be formatted. Please make appropriate changes. I would like to review these figures again once formatting has been completed.

Figure 2 has been reformatted and updated as outlined below:

Figure 2. Summary of methods used to inform and undertake formative research and intervention design.

Literature Review and Key Informant Interviews –

I would suggest dividing this section into 2 sections. In addition to this, it is important to provide information about questions that were used for this interview. Did authors use semi structured interview guide?

We have provided more detail on the structure of the stakeholder meetings, and although we are proposing to maintain the joint heading of ‘literature review and key informant interviews’ have broken the section into 2 paragraphs for ease of reading.

“A literature review was conducted to fully understand preceding food hygiene studies, their theoretical basis, methodologies and relative outputs in term of health, behavioral and food quality indicators. This process identified key studies and methods from which to build the formative research methodologies25,32,34,36,42,43, and an overview of the policy landscape within the country at that time. The findings of the literature review were used in the development of an effective formative research stage, and the development of a more detailed stakeholder analysis used to inform country and district specific information. 

The national and district stakeholder analysis was conducted through a national meeting (n=65 participants), a district executive committee meeting (n=25 participants), and key informant interviews with district and non-governmental staff (n=7) working in the study area, and community leaders (n=30). These face to face meetings were conducted as public discussion forums (national and district level), and focus group discussions (district and community level). Participants were presented with the findings of the literature review and an overview of the proposed research, which they were then asked to comment on based on previous experience and current knowledge. Information was also gathered on current programs and organizations working in the same geographical or subject areas which could be followed up. With specific reference to hygiene of complementary foods, the stakeholder analysis highlighted the issue of food safety as an ‘implied concept’ within WASH and nutrition programs, rather than explicit in its content and delivery.  It also underlined the challenges in prioritizing diarrheal disease due to the lack of specific policy or strategy addressing this disease44. Lastly, concerns were raised regarding the sustainability of interventions in the WASH sector, and how a specific program should ensure that sustained behavior change is at the core to improve health and well-being of the target population.

Section 2.6 – Please include information about “structured interview”. Again, I would like to see more information about questions that were asked during interview.

In section 2.6 we do not refer to ‘structured interviews’ but rather ‘structured observations’. We appreciate you may be referring to the use of IDIs and have added further clarification to the existing sentence:

“In-depth interviews followed each checklist and structured observation period to understand how and why some practices were conducted as observed”

I would also like to see more information about how data analysis was conducted for interviews noted above (see 1 & 2). Did you conduct thematic analysis, content analysis or phenomenological hermeneutics ??

Thank you for highlighting this oversight. The paragraph has been edited as follows:

“The national and district stakeholder analysis was conducted through a national meeting (n=65 participants), a district executive committee meeting (n=25 participants), and key informant interviews with district and non-governmental staff (n=7) working in the study area, and community leaders (n=30). These face to face meetings were conducted as public discussion forums (national and district level), and focus group discussions (district and community level). Participants were presented with the findings of the literature review and an overview of the proposed research, which they were then asked to comment on based on previous experience and current knowledge. Information was also gathered on current programs and organizations working in the same geographical or subject areas which could be followed up. All dialogue from the stakeholder meetings was subject to thematic analysis, and with specific reference to hygiene of complementary foods, highlighted the issue of food safety as an ‘implied concept’ within WASH and nutrition programs, rather than explicit in its content and delivery.  It also underlined the challenges in prioritizing diarrheal disease due to the lack of specific policy or strategy addressing this disease44. Lastly, concerns were raised regarding the sustainability of interventions in the WASH sector, and how a specific program should ensure that sustained behavior change is at the core to improve health and well-being of the target population.”

“2.6. Observations

For the identification of critical control points, checklist and structured observations were used, followed by in-depth interviews. Initially, checklist observations were conducted in 30 randomly selected households in formative research households to provide a list of behaviors that were considered to put children at risk of developing diarrhea. For the checklist observations, a household was visited over two consecutive days: 6 am – 12 noon on the first day and 12 noon to 6 pm on the second. The aim was to capture all events of interest that occurred in a day. Subsequently, structured observations were conducted, specifically focusing on behaviors noted during checklist observations. In total, 79 households were targeted for structured observations and they were visited once from 7 am to 1 pm, as checklist observations had informed that the majority of food preparation and feeding events took place in the morning. In-depth interviews followed each checklist and structured observation period to understand how and why some practices were conducted as observed. Data was analyzed using content analysis, and common risky practices noted during checklist observations that were further observed during structured observations were: child defecation; adult defecation; hand washing (after latrine use, after cleaning child’s bottom, before food preparation, before child feeding/eating, before breastfeeding); water (source management, collection, use and storage); animals and their feces in the compound; purchase, storage and consumption of raw food (including fruits); preparation and storage of cooked food; reheating of left-over food; washing and storage of utensils and child feeding. Upon further analysis of structured observation data, the following practices were selected as critical areas for control:” 

Other sections are well developed.

Many thanks